# Effects of Near-Infrared Light on Well-Being and Health in Human Subjects with Mild Sleep-Related Complaints: A Double-Blind, Randomized, Placebo-Controlled Study

**DOI:** 10.3390/biology12010060

**Published:** 2022-12-29

**Authors:** Marina Cecilia Giménez, Michelle Luxwolda, Eila G. Van Stipriaan, Pauline P. Bollen, Rieks L. Hoekman, Marthe A. Koopmans, Praveen R. Arany, Michael R. Krames, Anne C. Berends, Roelof A. Hut, Marijke C. M. Gordijn

**Affiliations:** 1Chrono@Work B.V., 9743 AD Groningen, The Netherlands; 2Groningen Institute for Evolutionary Life Sciences, University of Groningen, 9747 AG Groningen, The Netherlands; 3Department of Biology, Utrecht University, 3584 CS Utrecht, The Netherlands; 4Department of Oral Biology, Surgery and Biomedical Engineering, University at Buffalo, Buffalo, NY 14214, USA; 5Seaborough Life Science, 1098 XG Amsterdam, The Netherlands; 6Arkesso LLC, Palo Alto, CA 94306, USA

**Keywords:** photobiomodulation, near-infrared, sleep, human clinical trial, immune system, heart rate, mood, lighting

## Abstract

**Simple Summary:**

In Western societies, people spend most of their waking hours indoors, exposing themselves to virtually no sunlight. Natural sunlight contains all visible and non-visible spectral characteristics of light. Both play key roles in human health and well-being. In this particular context, the non-visible near-infrared light has been shown to be beneficial for a wide range of conditions. In the present study, we investigated the effects of morning exposure to near-infrared light five days per week for four consecutive weeks in a group (n = 56) of healthy individuals with mild sleep complaints. We observed consistent positive effects on several aspects of well-being and health but not on sleep or circadian rhythms. The benefits were only visible in the winter months, when sufficient exposure to sunlight is more challenging. The present study investigated rather low-energy light levels, which would allow for relatively easy incorporation of such technology into a household or personal appliances. Because of people’s indoor lifestyle and the need for more healthy buildings, the current results may open new ways of creating an optimal environment for a healthier society by preventing some negative effects produced by the lack of sunlight.

**Abstract:**

Modern urban human activities are largely restricted to the indoors, deprived of direct sunlight containing visible and near-infrared (NIR) wavelengths at high irradiance levels. Therapeutic exposure to doses of red and NIR, known as photobiomodulation (PBM), has been effective for a broad range of conditions. In a double-blind, randomized, placebo-controlled study, we aimed to assess the effects of a PBM home set-up on various aspects of well-being, health, sleep, and circadian rhythms in healthy human subjects with mild sleep complaints. The effects of three NIR light (850 nm) doses (1, 4, or 6.5 J·cm^−2^) were examined against the placebo. Exposure was presented five days per week between 9:30 am and 12:30 pm for four consecutive weeks. The study was conducted in both summer and winter to include seasonal variation. The results showed PBM treatment only at 6.5 J·cm^−2^ to have consistent positive benefits on well-being and health, specifically improving mood, reducing drowsiness, reducing IFN-γ, and resting heart rate. This was only observed in winter. No significant effects on sleep or circadian rhythms were noted. This study provides further evidence that adequate exposure to NIR, especially during low sunlight conditions, such as in the winter, can be beneficial for human health and wellness.

## 1. Introduction

In Western societies, people spend about 85% of their waking hours indoors [1], depriving themselves of exposure to direct sunlight. In temperate regions, outdoor sunlight can reach over 100,000 lux on a cloudless midday. Natural sunlight contains a broad spectrum of wavelengths ranging from far infrared (>2000 nm) to ultraviolet (UV 280–400 nm). On the other hand, indoor light intensities barely reach 500 lux and often contain only wavelengths within the visible range (420–740 nm). Moreover, modern window glazing (low-e glass coating) effectively blocks all wavelengths outside the visible range, especially NIR light, to improve building insulation.

Light has key roles in human health and wellness. Besides its role in enabling human vision, a direct role of UV-B (280–320 nm) in mediating vitamin D metabolism to maintain bones and teeth, as well as regulating inflammation and immune functions, has been well established [2,3]. Exposure to visible light is also known to affect sleep-wake rhythms, sleep quality, alertness, mood, and performance [4]. It has been reported that near-infrared (NIR 750–1100 nm) light accounts for about 54% of the solar radiation reaching the earth and is assumed to play an essential role in sustaining life on our planet [5], for instance, by improving health [6]. Studies on the exposure to red and NIR light, termed photobiomodulation (PBM) therapy, can be traced back to the 1960s, when its benefits on wound healing and the stimulation of hair growth were first observed [7]. Since then, PBM therapy has been reported to be effective for the treatment of a variety of conditions, such as wound healing, reducing inflammation or pain, and even treating depression [6,8,9,10]. In 2002, the U.S. FDA approved PBM treatment for pain relief in cases of head and neck pain, arthritis, and carpal tunnel syndrome [11]. More recently, PBM therapy has been recommended by the WALT/MASCC/ISOO guidelines as a standard of care for managing oral mucositis following oncotherapy such as chemo, radiation, or transplants [12,13,14].

The effectiveness of PBM is hypothesized to be related to the wavelength, dose, and pulse characteristics following a biphasic dose-response curve [15]. PBM is a non-thermal process involving endogenous chromophores eliciting photophysical (i.e., linear and nonlinear) and photochemical events at various physiological levels [8]. The most popular mechanism for PBM effects involves its actions on mitochondrial metabolism, cell membrane photoreceptors or transporters, and activation of extracellular latent growth factor, TGF-β1 [9,10,15,16,17,18,19,20,21,22,23,24,25]. Most applications of PBM focus on the local effects on tissue that is directly exposed to far-red or NIR light. However, indirect systemic effects of PBM on wound-healing, lung inflammation, and Parkinson’s disease have been demonstrated [18,26,27,28]. The deeper skin penetration of NIR compared to other visible light (red or blue) might contribute to the broader beneficial responses [29]. Chronic sleep deficits and/or circadian misalignment have been hypothesized to contribute to mitochondrial dysfunction [30,31]. These disruptions have been correlated with an increased risk for cardiovascular problems and a dysregulated immune system [32,33,34,35]. Given the effects of PBM on stimulating mitochondrial function, its potential therapeutic roles via systemic effects could benefit people with mitochondrial dysfunction afflicted with a broad range of diseases.

Decades of developments in the LED industry have finally provided NIR-LEDs with the adequate power and energy efficiency to provide adequate daylight-like exposure within standard indoor lighting infrastructure (see patent in the patent section). This study exploits these developments to explore the systemic effects of exposure to 850 nm NIR light in people suffering from mild sleep problems (i.e., sleep deficit and/or reduced sleep quality with clear complaints during daytime) in a double-blind, placebo-controlled study. Several outcomes within three categories were explored: health, well-being, and sleep. We hypothesized that PBM would have a positive effect compared to the placebo treatment and investigated the lowest effective dose in routine lighting for optimal therapeutic benefits.

## 2. Methods

### 2.1. Clinical Study Design

A double-blind, placebo-controlled study was conducted at participant’s homes. Each participant was allocated by stratified randomization to one of four conditions: placebo, low, medium, or a high dose based on age, chronotype, sleep quality, drowsiness, depression, sleep duration, and sleep deficit. NIR illumination of the skin of the face, neck, and hands was performed on subjects seated behind a desk. The study took place over three periods of the year, evenly distributed over summer and winter: January 2021–March 2021 (winter group), halfway April–July 2021 (summer group), and halfway November–December 2021 (winter group). Participants were recruited via advertisements and flyers. Inclusion criteria were: an average sleep duration per week ≤ 6.5 h, [36], and/or showing an accumulated sleep deficit of at least 1 h during the week [37], and/or reduced sleep quality (Pittsburgh Sleep Quality Index, PSQI > 5, [38]) with clear drowsiness complaints during daytime; Epworth Sleepiness Scale, ESS > 5 [39] or a mild depression score ≥ 13 and < 19; Beck’s Depression Inventory, BDI [40]. Exclusion criteria were: BDI > 19, pregnancy, menopause complaints, use of immunosuppressants, shift work, travel over more than 2 time zones, high alcohol intake (more than 4 units for men and more than 3 units for women per day, for more than 5 days in the past month), use of medications that are known to interfere with sleep, alertness, the biological clock and/or light sensitivity, and high levels of caffeine intake (5 or more cups per day). The study took place in the participant’s home and/or workplace for 4 consecutive weeks, mainly during home-work regulations by the government to prevent the spread of COVID-19. All study procedures were approved by the Medical Ethical Research Committee of the University Medical Centre Groningen (NL74857.042.20), The Netherlands. The procedures are in accordance with the Declaration of Helsinki (2013) and registered at the Netherlands Trial Register (#NL8800). All participants gave written informed consent and received financial compensation for their participation.

### 2.2. PBM Treatment

#### 2.2.1. Device

The PBM module was incorporated into a regular desk lamp (IKEA Ypperlig) (Figure 1A). It consisted of a wooden box with high-power 850 nm Lumileds NIR-LEDs (L1I0-0850090000000, Schiphol, The Netherlands) with a beam angle of 90 degrees. It was oriented in order to assure a 1sr beam that covered the user’s face, neck, and hands on the desk, with tissue surface irradiance of 5 mW·cm^−2^. This resulted in an estimated total illuminated area of approximately 1850 cm^2^ and of 1600 cm^2^ for males and females, respectively, as calculated using https://msis.jsc.nasa.gov/ (accessed on 7 December 2022), volume 1, section 3. Subjects were instructed not to cover the skin of their face, neck, and hands and not to use any skin products prior to treatment.

The module was further equipped with a distance and presence sensor (VL53L1X-SATEL, STMicroelectronics, Geneva, Switzerland), that allowed for NIR intensity adjustment to maintain a peak irradiance of 5 mW/cm^2^ on the skin as the user leaned forward or backward. Furthermore, if subjects came too close to the device (<20 cm), the radiation was switched off for safety reasons and a green LED would signal that an error had occurred, in which case participants knew they had to adjust their position. At the back of the module there was an ambient vertical lux sensor (BH1750FVI, ROHM Co., Ltd., Kyoto, Japan).

#### 2.2.2. Dose

The LEDs for PBM in this study were operated in pulsed mode with their corresponding duty cycle [15,41,42]. The PBM dose was established by changing frequency/duty cycle and the duration of the PBM resulting in doses of 0 J·cm^−2^, 1 J·cm^−2^, 4 J·cm^−2^, and 6.5 J·cm^−2^ (Table 1, Figure 1B). Other SI units often used to describe PBM dose are shown in Table 1. Dose and timing were programmed into the device, so that no user intervention was necessary. The NIR stimulation was not visible to the eye nor was it felt on the skin as heat. Strips of red (633 nm) LEDs (OSRAM, LS R976-NR-1, Munich, Germany) were blinking in low intensity at 10 Hz frequency to further prevent the possibility of visually noticing the NIR stimulation and to provide some user feedback regarding the device being “on’’.

Participants received a PBM module and were asked to sit in front of it from 9:30 am until 12:30 pm 5 times per week for four consecutive weeks. The PBM module switched on (9:30 am) and off (12:30 pm) automatically.

### 2.3. Outcomes Assessment

Actigraphy measurements (Motionwatch 8, Camntech, Fenstanton, UK, and Fitbit Versa 3, Fitbit Inc., San Francisco, CA, USA) started one day before the first PBM session. During the evenings of the baseline measurements scheduled 3 days prior to the first PBM session, as well as after the first and second session (day 1 and 2) and after 10 and 20 PBM sessions (week 2 and 4), the participants performed the following actions (Figure 2): filled out questionnaires regarding mood (1−10 Likert scale) and drowsiness (0−24 Epworth Sleepiness Scale (ESS, [39]); collected hourly saliva samples from 5 h before habitual sleep onset until 1 h after for assessment of Dim Light Melatonin Onset (DLMO) and cortisol; collected urine between 8 pm (after emptying the bladder) and 8 am for assessment of total night-time melatonin production. Furthermore, during the first 2 PBM sessions skin temperature was measured using iButtons.

Only at baseline and after 2 and 4 weeks, the following additional assessments were performed in the evening: questionnaires regarding sleepiness (1−9 scale, Karolinska Sleepiness Scale, KSS, [43]), subjective sleep quality (0−10 Likert scale), sleep complaints (0−21 scale, Pittsburgh Sleep Quality index, PSQI, [38]), WHO subjective performance (0−10 scale), and need for recovery (0−100 scale, [44]) were filled out. At daytime, blood was collected for the assessment of interferon gamma (IFN-γ) and tumor necrosis factor alpha (TNF-α).

Three assessments were only performed at baseline and after 4 weeks: blood collection for the assessment of vitamin D as an indication of exposure to outdoor light, hair sample collection for assessment of accumulated cortisol, and completing the BDI questionnaire.

#### 2.3.1. Saliva Collection

Participants were carefully instructed about the requirements for collecting saliva. No chocolate, bananas, artificially colored sweets, coffee, or black tea were allowed as well as brushing teeth with toothpaste. Eating and drinking were not allowed in the 30 min before the collection of saliva, and max. 10 min before each sample subjects were instructed to rinse their mouths with water. Subjects were also asked to expose themselves to as little light as possible by keeping the curtains closed, using only small light bulbs, dim light, decreasing brightness, using blue-blocking filters on screens, and wearing sunglasses inside, commencing 1 h before the first sample. Watching TV was allowed at a distance of ≥ 2 m. Postural changes were not allowed during the 5 min period before and during saliva collection. Saliva was collected by the participants using Salivette^®^ (Sarstedt™ Ltd., Nümbrecht, Germany) and stored overnight at approximately 4 °C. Samples were collected within 2 days, the Salivettes were centrifuged, and the saliva was transferred to 2 mL Eppendorf tubes and stored at −80 °C.

#### 2.3.2. Dim Light Melatonin Onset Assessment

On completion of the study, a double-antibody radioimmunoassay (RIA) was performed to assess melatonin concentration levels (Direct Saliva Melatonin kit, NovoLytiX GmbH, Switzerland; intra-assay variation: 20.1% and 4.8%; inter-assay variation: 16.7% and 8.4% for low and high concentration samples, respectively). Dim light Melatonin Onset (DLMO) was assessed for the first time when melatonin concentrations exceeded the 3 pg/mL threshold upon linear interpolation of subsequent melatonin values.

#### 2.3.3. aMTs6 Assessment

Urine was collected to measure degradation of melatonin by the production of 6-sulfatoxymelatonin (aMTs6) throughout the night between 8 pm and 8 am. It was stored at approximately 4 °C. Within 2 days, the urine was received by the research institute. The total volume was measured, and samples were transferred to 2 mL Eppendorf tubes and stored at −80 °C. On completion of the study, an enzyme-linked immunosorbent assay (ELISA) was performed (6-Sulfatoxymelatonin kit, NovoLytiX GmbH, Switzerland; intra-assay variation: 9.7% and 5.3%; inter-assay variation: 15.3% and 8.4% for low and high concentration samples, respectively).

#### 2.3.4. Cortisol Assessment

Cortisol analysis was carried out using the same saliva samples as used for melatonin analysis. The samples collected 3 and 4 h before bedtime were pooled (3.5 h before bedtime), as well as the samples collected 1 h before and at bedtime (0.5 h before bedtime). On completion of the study, a double-antibody radioimmunoassay (RIA) was used to assess cortisol concentrations levels (CORT-CT2 radioimmunoassay kit, Cisbio Bioassays, France; intra-assay variation: 3.9% and 3.2%; inter-assay variation: 7.6% and 5.1% for low and high concentration samples, respectively). Hair cortisol was collected by cutting a pencil-sized hair strand from the scalp at the back of the head and was stored in tinfoil at room temperature. On completion of the study, an online-solid phase extraction (SPE) combined with a fully validated isotope dilution liquid chromatography tandem mass spectrometry (LC–MS/MS) was performed (lab-developed, Department of Laboratory Medicine, Special Chemistry at the University Medical Center Groningen, The Netherlands [45]; intra-assay variation: 9.3% and 4.3%; inter-assay variation: 6.1% and 6.0% for low and high concentration samples, respectively). The lower limit of quantitation for hair cortisol was 0.70 pg/mg hair.

#### 2.3.5. Cytokines and Vitamin D Assessments

Two blood samples of 4 mL each were collected in EDTA-coated tubes. Samples were centrifuged, and plasma supernatant was recovered and stored in Eppendorf tubes at −80 °C. On completion of the study, an enzyme-linked immunosorbent assay (ELISA) was performed to assess IFN-γ and TNF-α concentration levels (Quantikine Human IFN-γ Immunoassay kit, and Human TNF-α Immunoassay kit, Bio-Techne Ltd., Abingdon, UK; intra-assay variation: 4.6% and 2.0% for IFN-γ and 2.2% and 1.9% for TNF-α; inter-assay variation: 9.8% and 7.2% for IFN-γ and 6.7% and 6.2% for TNF-α for low and high concentration samples, respectively).

Plasma 25-hydroxyvitamin D3 levels were determined using isotope dilution–online-solid-phase extraction liquid chromatography–tandem mass spectrometry (ID-XLC–MS/MS) (lab-developed, Department of Laboratory Medicine, Special Chemistry at the University Medical Center Groningen, The Netherlands; inter-assay variation: 4.5% and 6.8% for low and high concentration samples, respectively).

#### 2.3.6. Skin Temperature

Skin temperature was measured using iButtons (DS1922L, resolution: 0.0625 °C, Maxim Integrated, San Jose, CA, USA) on the first two days of PBM exposure. Participants were instructed to apply 7 iButtons: one on the forehead, two on the dorsal side of both middle fingers, 2 on the distal side of both claviculae, and two on the inner side of both ankles, and secured with Fixomull Stretch tape (BSN Medical B.V., Almere, The Netherlands). The iButtons were worn during the 3.5 h of PBM exposure + time needed for questionnaires (09:15 u–12:45 u). The same iButton was used on the same skin location for two consecutive days. Data were collected after the second PBM session using OneWire Viewer software.

#### 2.3.7. Composite Scores for Well-Being, Health, and Sleep Quality

Composite scores were calculated only from those outputs that were obtained after 2 and 4 weeks. Outputs were classified into three categories, and a composite score was calculated. The categories were: (1) well-being, including all questionnaire-related outputs, (2) health, including immune outputs, cortisol at bedtime, and resting heart rate (Fitbit), and 3) sleep quality, including actigraphy-derived sleep fragmentation, PSQI, the general sleep score, and minutes of deep sleep (Fitbit). Although the accuracy of the Fitbit-derived sleep parameters compared to polysomnography is still under debate, the within-subject changes are of interest for the current study [46].

To calculate the composite scores, each variable was Z-transformed, including all individual values at baseline, week 2, and week 4. For each subject at each timepoint, the 3 composite scores were calculated as follows: (1) the well-being composite score was the sum of each individual’s Z-transformed value of mood, drowsiness, sleepiness, need for recovery, and subjective performance, with each value being positive for better mood, less drowsy, less sleepy, less need for recovery and better performance; (2) the health composite score was the sum of each individual’s Z-transformed value of TNF-ɑ, IFN-γ, cortisol at bedtime (selected in view of a possible relationship with stress at bedtime), and resting heart rate value, again with a positive score meaning good health: less TNF-ɑ, less IFN-γ, lower cortisol at bedtime and lower resting heart rate; (3) the sleep quality composite score was the sum of each Z-transformed value of PSQI, sleep fragmentation, minutes of deep sleep and sleep score, with a higher sleep quality score meaning good sleep: high sleep quality, low PSQI, low sleep fragmentation score on the actigraphy, high amount of deep sleep minutes and high general sleep score on the Fitbit.

### 2.4. Data analysis

All statistics were performed in R (R Core Team, 2021; version: 4.1.0), using the most recent shell of R studio (version: 2022.02.0). ANOVA or T-tests were used to assess differences between groups’ demographics, light exposure during PBM sessions, and skin temperature data. Light intensity values from the light sensor at the back of the PBM module were transformed to a log(10) scale and averaged for each participant throughout the PBM session. Skin temperature data were z-transformed to the mean and SD of all conditions and all individuals and averaged throughout the first 40 min after the start of the PBM treatment (i.e., maximum PBM duration in the 1 PBM condition) for each individual. In the Appendix A, figures for the main items of well-being, health, and sleep are shown with the raw data (Appendix A). Given the in-between subjects design, responses were normalized to each individual’s baseline. This was done by subtracting baseline values of each variable from their corresponding output after a given time; after 1 or 2 days (short time-frame), as well as after 2 and 4 weeks (long time-frame). The analysis followed the subsequent structure: (1) long time-frame data were assessed by means of a composite score, and (2) if a significant effect was observed, the single items contributing to the composite score were explored. For the composite score as well as the single items, the interaction effect between PBM dose and time (week 2 vs. week 4) was first assessed by means of a mixed ANOVA, with dose and season as between factors and time as within factor. If no significant interaction with time was observed, an accumulated average of the difference between week 2 and baseline and of week 4 and baseline was used for further analysis. A similar approach was taken for the analysis of the short time-frame data (day 1 vs. day 2). No composite scores were calculated for the short time-frame data. Outputs are expressed as main effects of the factor PBM dose compared to placebo, as well as the factor PBM dose for the winter and summer groups, and the interaction effect between PBM dose and season. Motionwatch as well as Fitbit data were collected on a daily basis. For these outputs, an overall effect throughout all days as well as a delta for the first and second day relative to baseline were analyzed. Based on our a priori hypothesis and that the treatment (PBM dose) has more than 2 levels, the effects of dose and the potential modifying effects of season were tested as treatment contrasts by means of linear models. Linear models allow for the use of all data available per group as well as for construction of mixed-effects models. For the linear models, factors included were dose with 4 levels, season with 2 levels, and the interaction between dose and season. As secondary analysis, the effects of adding BMI and age as factors to the model and their interaction with dose were studied. Only significant interactions with BMI and age are reported. For the daily outputs of Motionwatch and Fitbit, the model also included days as a factor and its interaction with dose. Only the days in which the PBM module was used were included. Critical two-sided significance level alpha was 0.05 for all statistical tests. Due to the limited number of subjects per group (8 or less per season, Table 2), trends with an alpha up to 0.1 are also reported.

## 3. Results

Out of originally 62 subjects who were selected to participate, 56 completed the study (22 males and 34 females, age: 25−64, Caucasian: 51, non-Caucasian, non-Asian: 4, Asian: 2). Five participants cancelled their participation in an early stage, one due to illness not related to the study, one because of a pregnancy, and three because of personal circumstances. One participant was excluded from the analysis due to non-compliance. No significant differences between groups were observed, indicating a good a priori matching (Table 2).

To examine any variances in the groups between seasons, the winter and summer groups were assessed separately as well (Appendix A). No significant differences in any of the parameters were observed between the groups per season. A significant difference in chronotype and sleep deficit in the groups between seasons were noted. The winter season group showed an earlier chronotype (3.9 h ± 0.2 SEM) and less sleep deficit (0.9 h ± 0.1 SEM) than the summer season group (4.6 h ± 0.2 and 1.9 h ± 0.3, for chronotype and sleep deficit, respectively).

Analysis of the ambient light levels indicated that environmental indoor light levels (average 1.82 ± 0.56 SD log lux) did not differ significantly among the groups (F_3,48_ = 0.29, *p* = 0.83) despite the evident seasonal variation (1.96 ± 0.56 SD log lux summer and 1.69 ± 0.54 SD log lux winter). There was no significant interaction effect of the light intensity between the PBM dose and season (F_3,48_ = 0.2, *p* = 0.89), suggesting that PBM treatment effects cannot be explained by differences in environmental light exposure during the PBM sessions (Appendix A). These findings were further expanded to include outdoor light exposure, as shown by the changes in vitamin D throughout the study. While there was a significant effect of the season with a larger increase of vitamin D levels during the 4 week period in the summer (7.8 ± 11.2 SD) compared to a decrease (−4.2 ± 8.5 SD) in the winter (F_1,48_ = 18.96, *p* < 0.001), there were no significant differences among groups (F_3,48_ = 0.89, *p* = 0.45), nor was there a significant interaction between season and NIR dose (F_3,48_ = 0.56, *p* = 0.64; Appendix A).

### 3.1. Effects of PBM Treatment on Well-Being

For the composite score of well-being, no significant interaction between The PBM dose and time was observed (F_3,48_ = 0.46, *p* = 0.71). The cumulative average change in well-being over 2 and 4 weeks showed no significant effect of PBM treatment (−0.15 ± 1.15, −0.87 ± 1.13, 0.94 ± 1.15, for the 1, 4, and 6.5 PBM doses, respectively) compared to the placebo (1.98 ± 0.83, all *p* > 0.4). However, the analysis of the season factor revealed for the winter group a significant improvement in the 6.5 PBM group (3.65 ± 1.41, *p* < 0.01) compared to the placebo (0.46 ± 1.03).

Conversely, the summer group did not demonstrate any significant effect of the PBM treatments (−0.44 ± 1.52, −1.60 ± 1.51, −2.41 ± 1.57, for the 1, 4 and 6.5 PBM doses, respectively) compared to the placebo (3.77 ± 1.1 all *p* > 0.2). This is further supported by the significant interaction effect between the 6.5 PBM dose and the seasons (*p* < 0.001), indicating that the beneficial effects of PBM treatment on the participant’s well-being in the 6.5 J·cm^–2^ treatment condition are only present in the winter (Figure 3A, Appendix A). The main effect of the seasons was also significant, showing an overall increase of 3.30 points (± 1.52 SEM, *p* < 0.001) over the four weeks on the well-being score in the summer group, independent of PBM condition.

The individual items contributing to the composite well-being are elaborated below.

Mood: no significant interaction between the PBM dose and time was observed (F_3,48_ = 0.35, *p* = 0.89). The cumulative average over 2 and 4 weeks showed a tendency for an improvement of mood over time in the 6.5 PBM group of about half a point (±0.30 SEM, *p* < 0.1), compared to a smaller improvement in the placebo group (0.17 ± 0.22). The analysis of the season revealed in the winter group a significant improvement of mood over time in the 6.5 PBM group (1.46 ± 0.34, *p* < 0.001) compared to a small decrease in mood in the placebo group (−0.48 ± 0.25), which was also reflected in the significant interaction between the 6.5 PBM dose and season (*p* < 0.001). In summer, on the other hand, a small deterioration of mood over time was observed for the 4 PBM condition (−0.79 ± 0.37, *p* < 0.05) compared to a small improvement in the placebo group (0.94 ± 0.27) (Figure 4A, Appendix A). Overall, independent from condition, mood improved in the summer (1.4 ± 0.37 SEM) more than in the winter, as shown by the significant effect of the season factor (*p* < 0.001).

On a shorter time-frame (i.e., day 1 and day 2), a significant interaction effect between the factors PBM dose and time was found (F_1,48_ = 3.9, *p* < 0.05). Namely, a tendency for an improvement of mood over time was observed in the 6.5 PBM group (0.52 ± 0.31, *p* < 0.1), compared to the placebo group (−0.19 ± 0.22) after 1 day, while no effects of the PBM dose (−0.33 ± 0.31, −0.14 ± 0.31, 0.33 ± 0.31, for the 1, 4, and 6.5 PBM dose, respectively) compared to the placebo (0.29 ± 0.22, all *p* < 0.3) was observed on the second day. The analysis of the season factor revealed that these effects were only observed in the winter group, in which mood improved significantly over time in the 6.5 PBM group (1.01 ± 0.39, *p* < 0.05) after one PBM stimulation compared to the placebo group (−0.37 ± 0.29). A trend for a significant interaction between the 6.5 PBM dose and the season was observed (*p* < 0.1). No significant effects of the PBM treatment were observed after two stimulations. In the summer group, no effects of PBM treatment were observed for day 1 (0.61 ± 0.42, 0.53 ± 0.43, −0.09 ± 0.44, for 1, 4, and the 6.5 PBM dose, respectively) compared to the placebo group (−0.00 ± 0.31, all *p* > 0.2), or day 2 (−0.19 ± 0.45, −0.06 ± 0.45, −0.07 ± 0.47, for 1, 4, and 6.5 PBM dose, respectively), compared to the placebo group (0.48 ± 0.33, all *p* > 0.7).

Drowsiness: Examination of drowsiness ratings with the Epworth Sleepiness Scale revealed a significant interaction effect between the PBM dose, time (long-term), and season (F_3,48_ = 3.08, *p* < 0.05). Irrespective of the season, no effects of the PBM dose were observed after 2 weeks (−0.22 ± 0.95, 0.79 ± 0.94, −1.22 ± 0.96 for the 1, 4, and 6.5 PBM condition) compared to the placebo group (−0.92 ± 0.69, all *p* > 0.2), or after 4 weeks (−0.03 ± 1.19, 0.48 ± 1.17, −0.95 ± 1.19 for the 1, 4, and 6.5 PBM, respectively) compared to the placebo group (−1.61 ± 0.86, all *p* > 0.4). The analysis of the season factor revealed that it is only in the winter group that drowsiness is significantly reduced in the 6.5 PBM group (−2.57 ± 1.19, *p* < 0.05) over the first 2 weeks compared to the placebo group (−0.43 ± 0.87). A similar trend was noted after 4 weeks in the 6.5 PBM group (−2.98 ± 1.55, *p* < 0.1) compared to the placebo group (−0.14 ± 1.13). The cumulative average difference over 2 and 4 weeks showed a significantly larger reduction in drowsiness in the 6.5 PBM group in winter (−2.78 ± 1.25, *p* < 0.05) compared to the placebo control group (−0.28 ± 0.91). No effects of the PBM dose on drowsiness were observed in the summer group (−0.23 ± 1.34, 1.06 ± 1.34, 1.00 ± 1.39) compared to the placebo group (−2.41 ± 0.98, all *p* > 0.4) (Figure 4B, Appendix A).

Sleepiness: Examination of sleepiness ratings with the Karolinska Sleepiness Scale showed no significant interaction between the PBM dose and time (F_3,48_ = 0.69, *p* = 0.89). The cumulative average of sleepiness changes over 2 and 4 weeks showed no significant effect of the PBM dose compared to the placebo (all *p* > 0.3). The analysis of the season factor revealed no other significant effects of PBM treatment (Figure 4C, Appendix A).

On a shorter time-frame (i.e., day 1 and day 2) no significant interaction between the PBM dose and time was observed (F_3,47_ = 1.02, *p* = 0.35). The cumulative average change in sleepiness score over 1 and 2 days showed no significant effect of the PBM dose compared to the placebo (all *p* > 0.4). The analysis of seasons showed no other significant effects of PBM treatment (Appendix A). The KSS score was missing for one participant during the first evening in the placebo group.

Need for recovery: No significant interaction effect between the PBM dose and time was observed for the change in the need for recovery (F_3,48_ = 0.98, *p* = 0.41). The cumulative average over 2 and 4 weeks, irrespective of season, showed no significant effect of the PBM dose on the need for recovery compared to the placebo (all *p* > 0.7). The analysis of seasons showed no other significant effects of the PBM treatment on the need for recovery (Figure 4D, Appendix A).

The addition of BMI to the model revealed a significant effect in the cumulative average over 2 and 4 weeks of the 6.5 PBM dose; a larger increase in the need for recovery over time was observed (145.82 ± 68.81, *p* < 0.05) compared to the placebo (−105.4 ± 45.0), as well as an interaction effect between the 6.5 PBM dose and BMI (−5.97 ± 2.71, *p* < 0.05). This suggests that the higher the BMI, the less negative impact the 6.5 PBM dose has on the need for recovery. The analysis of the season factor showed a similar pattern. In the winter group, there is a tendency for a significant detrimental change in the need for recovery over time in the 6.5 PBM condition (141.84 ± 74.83, *p* < 0.1) compared to the placebo (−103.70 ± 52.97), as well as an interaction between 6.5 PBM and BMI (−6.09 ± 2.87), by which the effect is slightly compensated with a higher BMI. Furthermore, in the summer group, a similar observation was found; a significant increase in the need for recovery in the 6.5 PBM group (160.82 ± 71.81, *p* < 0.05) compared to the placebo group (−104.63 ± 47.25) as well as an interaction between 6.5 PBM and BMI (−6.09 ± 2.87) (Appendix A).

Subjective performance: No significant interaction effect between the PBM dose and time was observed in the subjective ratings of performance (F_3,48_ = 1.53, *p* = 0.22). The cumulative average over 2 and 4 weeks showed no significant effect of PBM treatment (0.09 ± 0.51, −0.11 ± 0.50, −0.36 ± 0.51, for 1, 4, and 6.5 PBM doses, respectively) compared to the placebo (0.61 ± 0.37, all *p* > 0.5). The analysis of the season factor showed in the summer, a tendency of a change in the subjective performance in the 6.5 PBM condition (−1.25 ± 0.73), compared to the placebo (0.83 ± 0.52, *p* < 0.1) (Figure 4E, Appendix A), suggesting a worsening of subjective performance. No effects of PBM treatment on subjective performance were observed in the winter group for any of the doses (0.21 ± 0.68, −0.68 ± 0.66, 0.32 ± 0.66 for the 1, 4, and 6.5 doses) compared to the placebo group (0.43 ± 0.48, all *p* > 0.3, Appendix A).

On a shorter time-frame (i.e., day 1 and day 2) no interaction effect between the PBM dose and time was observed in subjective performance ratings (F_3,47_ = 1.59, *p* = 0.34). The subjective performance score was missing for one participant during the first evening in the placebo group. Furthermore, the cumulative average over 1 and 2 days, irrespective of season, showed no significant effect of the PBM treatment on subjective performance ratings for any of the doses compared to the placebo (all *p* > 0.42). The analysis of the factor season revealed no other significant effects of the PBM treatment (Appendix A).

Depression: This outcome was not included in the composite score as it was only assessed at baseline and after 4 weeks (i.e., there was no 2 weeks measurement). No significant effect on the change of depression scores was observed for any PBM dose (0.19 ± 1.93, −0.17 ± 1.90, −0.59 ± 1.93) compared to the placebo (−3.69 ± 1.39, all *p* > 0.8). The analysis of the season factor showed a trend for an interaction effect between 6.5 PBM and season (−6.8 ± 3.86, *p* < 0.1), suggesting a larger reduction in depression scores in response to PBM treatment in the winter group (Appendix A).

### 3.2. Effects of PBM Treatment on Health

In the analysis of the composite score ‘health’, no interaction was found between the PBM dose and time (F_3,48_ = 0.85, *p* = 0.47). The cumulative average over 2 and 4 weeks showed a significantly larger improvement in health over time in the 6.5 PBM group (2.83 ± 0.95, *p* < 0.01) compared to the placebo group (−0.89 ± 0.68). The analysis of the season factor revealed that in the winter group, the 6.5 PBM dose asserts a significantly larger improvement over time (3.67 ± 1.26, *p* < 0.01) compared to the placebo group (−0.66 ± 0.9), but not in the summer group (1.68 ± 1.40) compared to the placebo group (−1.17 ± 0.99, *p* = 0.24) (Figure 3B, Appendix A). The changes in individual items are further elaborated below.

IFN-γ: No interaction was found between the PBM dose and time (F_3,48_ = 0.72, *p* = 0.55). The cumulative average over 2 and 4 weeks showed a significantly larger reduction in IFN-γ concentrations of about 1.7 pg/mL (± 0.71 SEM, *p* < 0.05) compared to the placebo (0.27 ± 0.51 pg/mL). The analysis of the season factor revealed only in the winter group a significantly larger reduction in the 6.5 PBM group of about 3 pg/mL (−2.84 ± 0.92 pg/mL, (*p* < 0.01), compared to a small increase in IFN-γ concentrations in the placebo group (0.49 ± 0.68 pg/mL) (Figure 5A, Appendix A). No effect of treatment was found in the 6.5 PBM condition in the summer group (−0.18 ± 1.03 pg/mL) compared to the placebo group (0.01 ± 0.73 pg/mL, *p* = 0.86).

TNF-α: No interaction was found between the PBM dose and time (F_3,48_ = 0.53, *p* = 0.66). The cumulative average over 2 and 4 weeks showed no effect of the PBM treatment on the TNF- α concentration compared to the placebo (all doses *p* > 0.2). The analysis of the season factor revealed no further effects of the PBM treatment in either the winter or summer group (Figure 5B, see Appendix A).

The addition of BMI to the model led to the observation of a significant reduction of about 1 pg/mL of TNF- α in the 6.5 PBM group over time (−1.09 ± 0.47 pg/mL, *p* < 0.05) compared to a small increase in the placebo group (0.40 ± 0.30 pg/mL), as well as to a significant interaction effect between BMI and the dose 6.5 PBM (0.04 ± 0.02 pg/mL, *p* < 0.05) (Appendix A). This means that higher BMI levels prevent to a small extent the decrease in TNF-α in response to the 6.5 PBM treatment. The analysis of the season factor revealed that the effect of the 6.5 PBM treatment was present in the winter group (−1.09 ± 0.49 pg/mL), but significantly different compared to the placebo group (0.41 ± 0.35 pg/mL, *p* < 0.05) as well as in the summer group (−1.16 ± 0.47 pg/mL, *p* < 0.05) compared to the placebo (0.41 ± 0.31 pg/mL). Again, an interaction between 6.5 PBM and BMI was observed (0.04 ± 0.02 pg/mL, *p* < 0.05), indicating that higher BMI levels reduce the reduction in TNF-α levels in response to PBM treatment.

Cortisol 0.5 h before bedtime: No interaction effect between the PBM dose and time was found (F_3,46_ = 0.68, *p* = 0.57). The cumulative average over 2 and 4 weeks showed a significant reduction in cortisol levels at bedtime in the 6.5 PBM condition (−8.05 ± 3.86 nmol/L, *p* < 0.05) compared to the placebo condition (3.60 ± 2.78 nmol/L). The analysis of the season factor revealed no further effects of the PBM treatment with any dose neither in the winter group (−2.79 ± 5.06, −1.92 ± 5.38, −8.25 ± 5.21 for the 1, 4, and 6.5 PBM doses, respectively) compared to the placebo group (0.79 ± 3.81 nmol/L, all *p* > 0.1), nor in the summer group (−7.46 ± 5.60, −3.90 ± 5.60, −7.31 ± 5.81 for the 1, 4, and 6.5 PBM doses, respectively) compared to the placebo group (6.87 ± 4.11 nmol/L, all *p* > 0.2) (Figure 5C, Appendix A). Two participants (1x 1PBM and 1x 4PBM condition) did not generate enough material for a cortisol analysis.

The analysis of the effect of the PBM dose on cortisol levels at bedtime on a shorter time-frame (i.e., day 1 and day 2) did not reveal any significant interaction with assessment time (F_3,48_ = 1.27, *p* = 0.29). The cumulative average over 1 and 2 days showed no significant effects for PBM compared to the placebo (all *p* > 0.2). The analysis of the season factor revealed no further effects of the PBM treatment (Appendix A).

Cortisol 3.5 h before bedtime: This was not included in the composite score as the largest expectations were focused on cortisol levels just prior to bedtime and the data points are closely related. Three participants (1 in 1PBM condition and 2 in 4 PBM condition) did not generate enough material for cortisol analysis. No interaction effect was observed between the PBM dose and time in analyzing the effect on evening cortisol levels (F_3,45_ = 0.64, *p* = 0.59). The cumulative average over 2 and 4 weeks showed no significant effects of PBM compared to placebo (all *p* > 0.3) on the change in the level of evening cortisol. The analysis of the season factor revealed no further effects of the PBM treatment in the winter summer groups. See Appendix A.

On a shorter time-frame (i.e., day 1 and day 2), the PBM dose and time showed no significant interaction effect (F_3,48_ = 0.32, *p* = 0.81) on the change in evening cortisol levels. The cumulative average over 1 and 2 days showed a tendency for a reduction in evening cortisol concentrations over time in the 6.5 PBM condition (−3.03 ± 1.81 nmol/L, *p* < 0.1) compared to the placebo (0.65 nmol/L ± 1.30). The analysis of the season factor revealed no effect of the PBM dose in the winter group (−0.29 ± 2.61, 0.20 ± 2.52, −2.74 ± 2.52 for the 1, 4, and 6.5 PMB doses, respectively) compared to the placebo group (0.43 ± 1.84 nmol/L, all *p* > 0.3), or in the summer group (−0.76 ± 2.71, −1.35 ± 2.71, −3.38 ± 2.82 for the 1, 4, and 6.5 PMB doses, respectively) compared to the placebo group (0.90 ± 1.99 nmol/L, all *p* > 0.2).

Cortisol in hair: This was not included in the overall health composite score as it was only assessed at baseline and after 4 weeks. For three participants (1 in 1 PBM, 1 in 4 PBM and 1 in 6.5 PBM) it was not possible to quantify cortisol in hair.

A significant increase over time in the amount of cortisol in the participant’s hair was observed for the 1 J·cm^–2^ PBM condition (2.19 ± 0.87 pg/mg, *p* < 0.05) compared to placebo (−0.75 ± 0.24 pg/mg). The analysis of the season factor revealed no effect of the PBM treatment at any dose on the changes of the accumulated cortisol concentration in the winter group (1.05 ± 1.21, 0.07 ± 1.21, −0.07 ± 1.21 for the PBM 1, PBM 4, and PBM 6.5 conditions, respectively) compared to the placebo (−0.16 ± 0.86 pg/mg, all *p* values > 0.4). However, the summer group showed a significant increase of accumulated cortisol levels in 1 J·cm^–2^ PBM dose (3.56 ± 1.32 pg/mg, *p* < 0.05) compared to placebo (−1.57 ± 1.01 pg/mg).

Resting heart rate (RHR): Resting heart rate was assessed continuously using a wearable sensor throughout the period of the study. To include this parameter in the health composite score, the RHR assessed at night after 2 and 4 weeks was compared to the RHR at the baseline night in a similar way as with the previous parameters. For two participants (i.e., 1 in in the placebo group and 1 in in 6.5 PBM treatment group), it was not possible to measure RHR, and one participant in the 4 PBM group did not wear the device. No interaction effect was observed between the PBM dose and time (F_3,45_ = 1.54, *p* = 0.22). The cumulative average over 2 and 4 weeks showed no significant effects of PBM treatment (−0.66 ± 1.39, −0.79 ± 1.39, −2.35 ± 1.41, for 1, 4, and 6.5 PBM, respectively) compared to the placebo (1.03 ± 1.02, all *p* > 0.1). When accounting for the season, the analysis showed that the RHR was significantly reduced in the winter group with 6.5 PBM treatments (−4.60 ± 1.90 bpm, *p* < 0.05) compared to the placebo (1.85 ± 1.34 bpm) but only a tendency in the summer group (0.45 ± 2.15 bpm versus placebo −0.11 ± 1.59 bpm, *p* = 0.08). See Figure 5D, Appendix A.

Analyzing the daily pattern of the RHR over the whole four-week period, it was noted that the changes in the RHR were significantly lower on days that the PBM module was used in the 4 PBM group (−2.73 ± 0.99 bpm, *p* < 0.01) compared to the placebo (64.26 ± 0.74 bpm). Further analyses of the seasonal factor showed that these effects were significant in the winter group for both the 4 PBM (−5.7 ± 1.61 bpm, *p* < 0.001) and the 6.5 PBM dose (−4.1 ± 1.65 bpm, *p* < 0.05) compared to the placebo group (66.5 ± 1.19 bpm). A significant interaction effect with season was observed for the 4 PBM dose (*p* < 0.05) and only a tendency with the 6.5 PBM one (*p* < 0.1), indicating that the effects were not present in the summer group (Figure 6A,B). The factor days was not significant (*p* = 0.7) meaning that RHR remained at a given level throughout the four weeks. The lowering of RHR appeared to be maintained on days that the PBM module was not used PBM use (y/n) was not significant (*p* = 0.8) nor was its interaction with the PBM dose (all *p* > 0.2).

On a shorter time-frame (i.e., day 1 and day 2) the PBM dose did not reveal a significant interaction with time (F_3,44_ = 1.97, *p* = 0.13). The cumulative average over 1 and 2 days showed a significant reduction of treatment on the RHR in both the 4 PBM group (−1.79 ± 0.6 bpm, *p* < 0.01) and 6.5 PBM group (−1.84 ± 0.61 bpm, *p* < 0.01) compared to the placebo (0.76 ± 0.45 bpm). Consistent with other results, significant changes in RHR were only present in the winter group in the 4 PBM (−2.26 ± 0.8 bpm, *p* < 0.01) and the 6.5 PBM group (−2.57 ± 0.83 bpm, *p* < 0.01) compared to the placebo group (1.47 ± 0.61 bpm) (Figure 7). A trend for a reduction was also observed in the one PBM condition (−1.61 ± 0.83 bpm, *p* = 0.06). No significant effects of PBM treatment were observed in the summer group (−0.63 ± 0.87, −1.27 ± 0.90, −0.96 ± 0.9, for the 1, 4, and 6.5 doses, respectively) compared to the placebo group (−0.09 ± 0.67 bpm, all *p* > 0.2). For four participants (i.e., 2 in placebo, 1 in 4 PBM and 1 in 6.5 PBM) RHR data were not available for one of the nights, and therefore they were excluded from the analysis.

### 3.3. Effects of PBM Treatment on Sleep Quality

The analysis of the effect of PBM treatment on the composite score of sleep quality revealed no significant interaction between the PBM dose and time (F_3,48_ = 1.53, *p* = 0.22). The cumulative average over 2 and 4 weeks showed no significant effect of PBM treatment at any dose on sleep quality compared to the placebo group (all *p* > 0.1). The analysis of the season factor revealed no further effects of the PBM treatment, neither in the winter nor in the summer group (Figure 3C, Appendix A).

### 3.4. Effects of PBM Treatment on Circadian Rhythm

Dim Light Melatonin Onset (DLMO: PBM treatment on DLMO revealed no significant interaction between time and dose (F_3,33_ = 0.83, *p* = 0.46). The cumulative average over 2 and 4 weeks showed no significant changes in the phase of the melatonin rhythms following PBM treatment for any dose (0.08 ± 0.37, −0.00 ± 0.39, 0.46 ± 0.39, for 1, 4, and 6.5 PBM doses, respectively) compared to the placebo (−0.48 ± 0.28, all *p* > 0.2).

In view of the number of participants for which the DLMO was not possible to calculate, the analysis of the season factor was not possible. A total sample size of 9, 12, 10, and 10 were available for the placebo, 1, 4, and 6.5 PBM, respectively. Out of the nine available in the placebo group, only three belong to the summer group.

On a shorter time-frame (i.e., day 1 and day 2), the PBM dose showed no significant interaction with time (F_3,36_ = 1.21, *p* = 0.32). The cumulative average over 1 and 2 days showed no significant effects of PBM treatment compared to the placebo group (-all *p* > 0.6). In view of the limited number of participants for whom it was possible to analyze DLMO, the analysis of the season factor is not possible. A total sample size of 8, 11, 13, and 12 were available for the placebo, 1, 4, and 6.5 PBM, respectively. Out of the eight available in the placebo group, only three belong to the summer group.

aMTs6: For five participants (i.e., 1 placebo, 1 in in 1 PBM group, 1 in in 4 PBM group and 1 in in 6.5 PBM group) it was not possible to quantify aMTs6. Further, one participant (6.5 PBM) did not collect night-time urine during the study. No significant interaction between the PBM dose and time (F_3,42_ = 0.46, *p* = 0.71) was observed. There was no indication of a change in the amount of melatonin degradation product produced at night following the PBM treatment compared to the placebo group (all *p* > 0.6). The analysis of the season factor revealed no further effects of the PBM treatment, neither in the winter nor in the summer group, see Appendix A.

On a shorter time-frame (i.e., day 1 and day 2), PBM treatment did not result in a significant interaction effect with the PBM dose and time (F_3,44_ = 1.35, *p* = 0.27). The cumulative average over 1 and 2 days showed no significant effects of PBM treatment on aMTs6 compared to the placebo group (all *p* > 0.4). The analysis of the season factor revealed no further effects of the PBM treatment, neither in the winter nor in the summer group, see Appendix A. For the short time-frame, three participants (i.e., 1 in placebo, 1 in 4 PBM and 1 in 6.5 PBM) it was not possible to quantify aMTs6. Further, one participant (6.5 PBM) did not collect night-time urine during the study.

### 3.5. Temperature

During PBM treatment, no significant changes in skin temperature for the 4 and 6.5 PBM groups compared to the placebo were found, irrespective of anatomical location (all *p* > 0.2). This indicates that there was no acute effect on the temperature or any indications for thermoregulatory process from the PBM treatments. For the one PBM dose, a significant increase in temperature (0.82 °C) at the head location only was observed (t = 2.64, *p* < 0.05, Appendix A).

## 4. Discussion

The goal of the present study was to assess whether a PBM set-up used at home with a treatment period of several hours per day during several weeks could be beneficial for general well-being, health, sleep quality, and circadian entrainment in healthy subjects with mild sleep-related complaints. The analysis of the composite scores showed that well-being and health were positively affected at the highest dose of PBM (6.5 J·cm^−2^) during winter, while estimates of sleep quality were not affected. Nor were any circadian-related outputs affected by PBM treatment. To our knowledge, this is the first time that the systemic effects of PBM have been studied in healthy subjects during their normal daily routine in a double-blind, randomized, placebo-controlled trial.

Near-infrared (NIR) stimulation is a natural phenomenon that occurs when humans are exposed to sunlight. The positive health effects of sunlight have often been attributed to the vitamin-D3, β-endorphin, serotonin production in response to UV-B exposure [47,48,49]. In a recent paper, Heiskanen et al. questioned the role of the effects of vitamin D3 alone and suggested that red and NIR light may also be a component of the positive effects on health [6]. It should be noted that indoor NIR irradiance from general lighting conditions is at least 100 times lower than direct sunlight. This is likely too low to induce an appreciable biological benefit, even with previously used incandescent light sources, which do emit NIR radiation. Achieving 500 lux using 3000 K incandescent lamps delivers an irradiance of 1.5 W·m^−2^ in the 800−900 nm spectral window. Even if the low irradiance of 1.5 W·m^−2^ would be sufficient to reach the PBM threshold, it would require almost 12 h exposure to achieve a 6.5 J·cm^−2^ PBM dose.

The 6.5 J·cm^−2^ PBM dose is a reasonable treatment similar to natural sunlight exposure. If people were outside and uncovered on a clear summer’s day in The Netherlands, we could expect an 800−900 nm irradiance of about 90 W.m^−2^ at midday [50]. The 6.5 J·cm^−2^ PBM condition used in the present study would then be achieved after about 12 min of natural light exposure. On a clear winter’s day, this would be achieved after about 16 min, while on a cloudy day one would need to spend about 7 h outdoors (energy of about 2.5 W·m^−2^ at midday for the 6.5 J·cm^−2^ PBM condition [50]. Considering that in the winter months people wear protective clothes with very little skin exposed to sunlight due to the weather conditions, and that sunny winter days are exceptional occasions in Northern Europe, the 6.5 J·cm^−2^ NIR exposure would rarely ever be reached. The ease with which a natural PBM dose can be experienced in summer might explain the consistency of our stand-out findings occurring only in the winter group, even though the sample size for the season comparison was relatively small and should be considered a limitation on which to improve in following studies.

In the present study, many parameters were analyzed. To gain statistical power, three main composite scores were created to test for overall effects on well-being (several subjective ratings), health (several physiological markers), and sleep (several subjective and objective estimates of the sleep-wake rhythm). A more detailed analysis of the well-being and health composite scores showed consistent results for various parameters. In the well-being category, drowsiness and mood mostly contributed to the positive PBM effects (Figure 4). This is strengthened by the consistent finding that mood was already improved by the highest PBM dose immediately after the first day and for the following four weeks. PBM, mostly transcranially, has repeatedly been shown to be able to positively influence mood in different sorts of conditions (e.g., brain injury, traumatic events, and depression, as well as in healthy subjects) [51]. PBM has also recently been discussed as a tool for the treatment of depression, but the authors do not outline optimal treatment parameters as the variation in these reported parameters is too large to be summarized [18,52]. Using the BDI as a scale, Henderson et al. found, unlike us, a reduction in depressed mood [53]. The study included participants who had a much higher BDI score at inclusion about 24 compared to our baseline value of about 11, and much higher NIR irradiances were used (13.2 W at 0.89 cm^2^, 810 nm). We explicitly excluded potential candidates with a BDI above 19, which has been linked to moderate depression [40], as we wanted to test the PBM intervention in healthy subjects. In addition, we aimed at identifying an effect with the lowest possible intensity. Nonetheless, we saw a trend for a reduction in depression in the 6.5 J·cm^−2^ PBM condition during the winter months (Appendix A). This interaction between PBM and the season was also observed for the well-being composite score. This could be explained by the fact that the participants were more likely to be exposed to natural sunlight during the summer months to such an extent that the additional PBM treatment had no effect on the subjective parameters contributing to the well-being composite score. To some extent, the increase in vitamin D levels in the summer group over the four weeks in all conditions confirms this. The positive effects on well-being being visible in winter, and significantly different from the placebo group strengthens the conclusion that it is the 6.5 J·cm^−2^ PBM treatment that is responsible for the positive effects. Furthermore, the well-known seasonal variation that some of the well-being items show, by which a decrement is observed in the winter months [54], might have contributed to the observed interaction, i.e., more room for improvement during the winter months.

The composite score for health also showed a positive effect in the 6.5 J·cm^−2^ PBM group. This is primarily due to a reduction in cytokine IFN-γ and a reduction in the resting heart rate (Figure 6). IFN-γ is key in driving inflammatory responses against both exogenous and endogenous species [55,56]. Downregulation of IFN-γ after exposure to far-red (670 nm) light has already been shown in mice after full body exposure [57] and in vitro [58]. In general, one of the most reproducible effects of PBM is a reduction in inflammation [59], but to our knowledge, this is the first study that shows such a systemic reduction in humans after a four-week treatment period. The positive effect of PBM in reducing inflammation has even resulted in a discussion as a potential treatment for COVID-19 [60,61,62]. An anecdotal additional finding to this discussion is that during the current study carried out in the year 2021, only 1 out of the 62 initially included participants (PBM condition 4 J·cm^−2^) tested positive during the four-week study. This participant showed only mild symptoms and recovered within two days. With a very high rate of infections this year, this is an intriguing finding. Future studies, testing specifically on markers for COVID-19, could provide further support for a possible protective role of PBM on the immune system. The cytokine TNF-α was significantly reduced in the 6.5 J·cm^−2^ dose group only when BMI was considered as part of the model (Appendix A). The interaction effect indicates that a higher BMI hampers to some extend the reduction of TNF-α by PBM. This effect could be explained by the different metabolism of fat cells in obese people compared to non-obese people. The production of cytokines amplifies with increasing BMI [63,64] and fat cells of obese people produce 5 to 10-fold higher TNF-α mRNA compared to lean fat cells [65,66]. It may be interesting to test people who are overweight and show high levels of inflammation for a longer period than 4 weeks and/or with higher doses of NIR.

The selection of this particular group of subjects was premised on the mitochondrial mechanism of PBM. This is the most accepted mechanism of action, and it is based on photons being absorbed by cytochromes, which are present inside the mitochondria either as functional proteins or as electron transport shuttles (i.e., cytochrome c oxidase, CCO). This leads to an increased production of ATP, increased oxygen consumption, raised mitochondrial membrane potential, and increased mitochondrial biogenesis, which have all been shown in vitro after PBM, resulting in transient increase in ATP, ROS, and NO levels [9,20,21,59,67]. Mitochondrial dysfunction is thought to be a causal factor in the detrimental health effects of sleep deficits, especially in people who are late chronotypes [30,31]. The reported potential role of NO having a positive effect on mood may be relevant as well. The positive effects of ambulatory PBM treatment during daytime hours, if effective via mitochondrial stimulation, may be particularly suitable for the general public suffering from sleeping problems. Although the positive health and well-being effects were indeed noted in the current study, they do not seem to be mediated by positive effects on the circadian rhythm or sleep. Further insight into the mechanisms underlying the effects is necessary. It is feasible that PBM may modulate immune cell functions by modulating their mitochondrial functions. This explanation has been hypothesized to be one of the mechanisms by which PBM could assert a systemic effect [68]. The positive effects of light on the skin and the role of humoral phototransduction in the effects on mood have been discussed in other contexts, such as light exposure during winter depression [69,70]. However, studies on light treatment at visual wavelengths through the skin were not found to be effective in treating mood or affecting the circadian system [71]. The evidence for direct modulation of the cytokines IFN-γ and TNF-α levels in this study raises interesting possibilities of other systemic factors that are being directly modulated by PBM treatments, as has been shown with TGF-β1 [72]. Moreover, several studies report that pro-inflammatory cytokines play an important role in the brain in the pathogenesis of mood disorders [73,74,75]. This study noted a significant reduction in cytokines levels and improved mood which may be an interesting aspect of future PBM studies, potentially providing mechanistic insights for therapeutic PBM responses.

Another parameter contributing to the composite score ‘health’ is the resting heart rate (RHR). The RHR represents the balance of sympathetic and parasympathetic activity and is considered a reliable marker of autonomic nervous system tone [76]. The association between an increased RHR and adverse health outcomes in the general population has been widely investigated [77]. We observed an immediate change in the RHR after the first PBM session (4 J·cm^−2^ and 6.5 J·cm^−2^ PBM conditions), that is maintained throughout the whole study (Figure 7). We included cortisol levels at bedtime as a fourth item in the composite score for health. Higher evening cortisol levels have been found in mood disorder patients [78,79,80] and may be related to higher stress and arousal levels. Higher bedtime levels of cortisol have also been reported to be related to disrupted sleep [81,82]. Although a reduction in cortisol at bedtime was observed after treatment with 6.5 J·cm^−2^, this was no longer visible when exploring the differences between the winter and summer groups. This could be attributed to lower statistical power. Cortisol earlier in the evening was not affected by PBM, suggesting that the effects are indeed more related to the ease of the moment just prior to falling asleep. The lack of a consistent effect of PBM on the accumulated cortisol level over 4 weeks as measured in hair samples does not support the idea that PBM reduces cortisol in general.

Although we did not specifically seek to identify side effects in this study, from our regular contact with the participants, a few negative experiences were exchanged. These included headaches, eyestrain, dizziness, tiredness, and dryness of the skin (Appendix A). As evident from the data, these reports constituted a very small number of participants (between 1 and 3) and did not correlate with the PBM dose specifically, suggesting that the way PBM treatment was delivered was largely acceptable.

## 5. Conclusions

The findings from the current study, that 6.5 J·cm^−2^ PBM treatment improves several health and well-being-related factors, are supported by previous reports on the beneficial effects of PBM. That our findings are consistent in the short and long term and only present in the winter strengthens our observations. Still, a replication of the present study in terms of its set-up (i.e., home, LED-based) is desired, as well as studies on possible cellular mechanisms and pathways involved in mediating the effects on health and well-being. Future prospective research on dose, duration, timing, and potential mechanisms is expected to further strengthen our conclusions.

The present study only investigated rather a low-energy PBM treatment, and the optimal dosage might not have been reached. While it is known that increasing energy could intensify the stimulation and the possible effects before reaching a plateau [15], the present positive findings at such low intensities suggest the possibility of the incorporation of PBM into household or personal appliances (i.e., considering energy costs). It seems important to intensify research on the effects of non-visual long wavelengths in addition to the important research on the effects of visual light on non-image-forming functions [83,84]. In light of our indoor lifestyle and the need for more healthy buildings, the current results may open a completely new way of creating an optimal environment for a healthier society.

## 6. Patents

The patent number of NIR-LEDs is WO2020119965A1, can be found at https://patents.google.com/patent/WO2020119965A1/en?oq=wo+2020%2F119965A1 (accessed on 7 November 2022).

## Figures and Tables

**Figure 1 biology-12-00060-f001:**
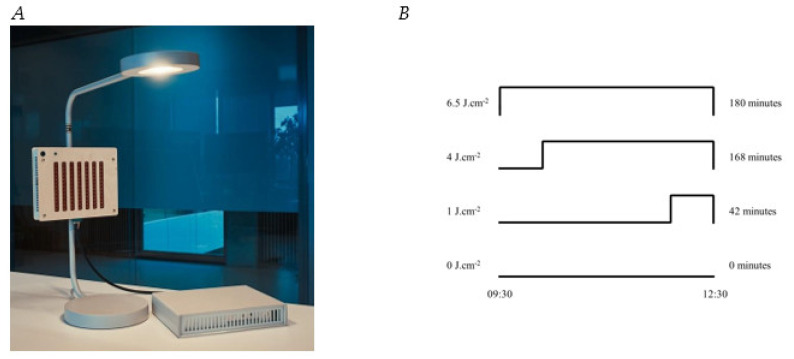
(**A**) PBM treatment module used in this study. The PBM module is incorporated in a desk lamp and its power is controlled by the driver at the table. The strips of low-power red LEDs at the front of the module were turned on in all conditions. The distance sensor (large circle) is shown. The smaller circle in the upper left corner is a green LED that provides feedback on the distance of the user. (**B**) A schematic overview of the different PBM dose durations all timed relative to the offset of the PBM session.

**Figure 2 biology-12-00060-f002:**
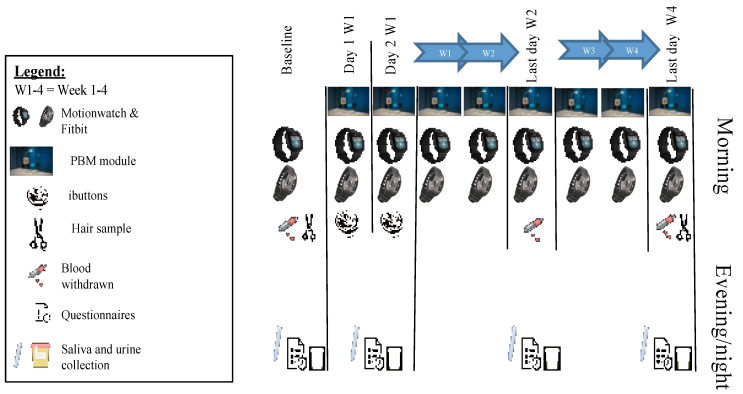
Schematic overview of the study design and performed measurements.

**Figure 3 biology-12-00060-f003:**
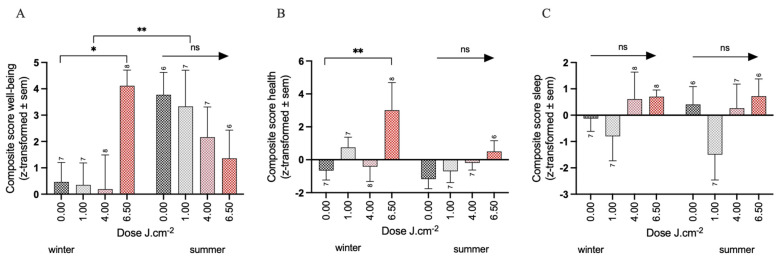
Change in composite score for winter and summer separately (average of the difference between week 2 and baseline and of week 4 and baseline) for (**A**) well-being, (**B**) health, and (**C**) sleep. Significance codes: ** *p* < 0.01, * *p* < 0.05, ns: not significant. Sample sizes per condition are shown.

**Figure 4 biology-12-00060-f004:**
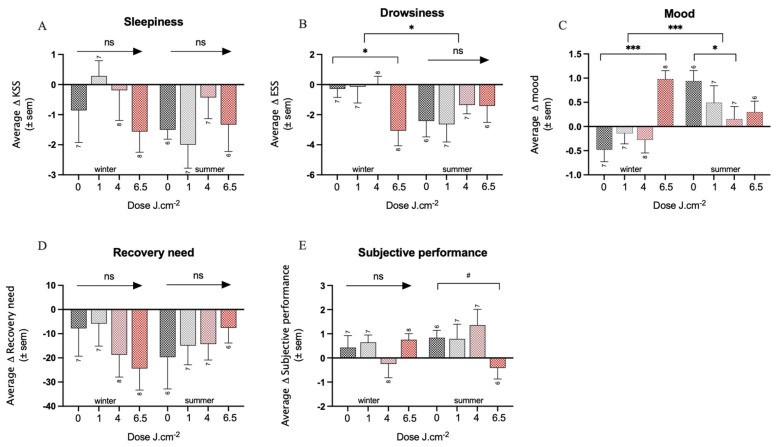
Overview of all individual elements of the well-being composite score, for the summer and winter separately. (**A**) mood, (**B**) drowsiness (ESS), (**C**) sleepiness (KSS), (**D**) need for recovery, and (**E**) subjective performance. Significance codes: *** *p* < 0.001, * *p* < 0.05, # *p* < 0.1 ns: not significant. Sample sizes per condition are shown.

**Figure 5 biology-12-00060-f005:**
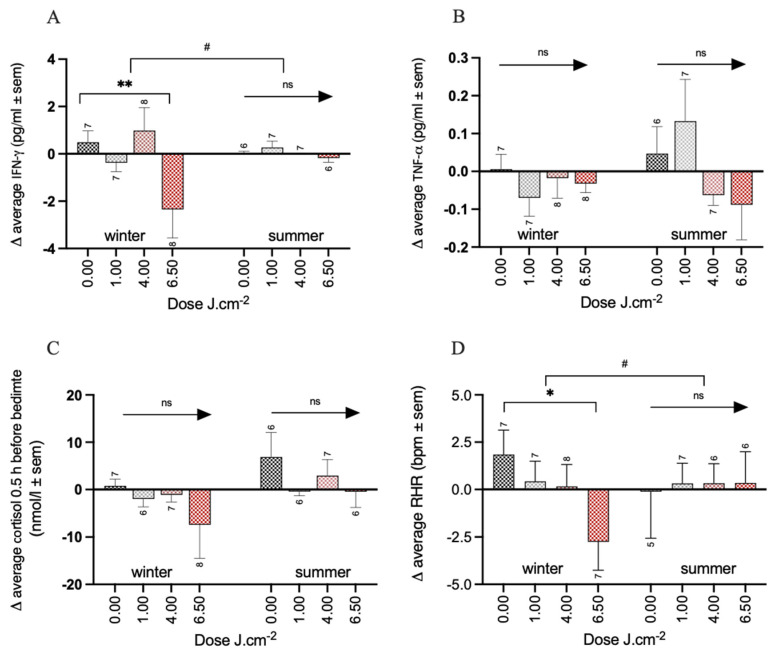
Overview of all individual elements of the health composite score for the summer and winter separately. (**A**) IFN-γ, (**B**) TNF-α, (**C**) cortisol 0.5 h before bedtime, and (**D**) RHR of the night after 2 and 4 weeks of PBM. Significance codes: ** *p* < 0.01, * *p* < 0.05, # *p* < 0.1 ns: not significant. Sample sizes per condition are shown.

**Figure 6 biology-12-00060-f006:**
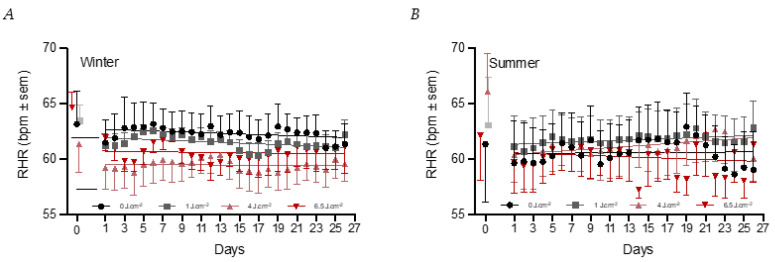
Daily RHR during the four week period. The baseline values are shown on the left side. The black circles represent the 0 J·cm^−2^ condition (n = 7:6 for winter and summer), grey squares the 1 J·cm^−2^ condition (n = 7:7 for winter and summer), pink triangles the 4 J·cm^–2^ condition (n = 8:7 for winter and summer), and red triangles the 6 J·cm^–2^ condition (n = 8:6 for winter and summer). (**A**) shows the data in the winter group and (**B**) of the summer group.

**Figure 7 biology-12-00060-f007:**
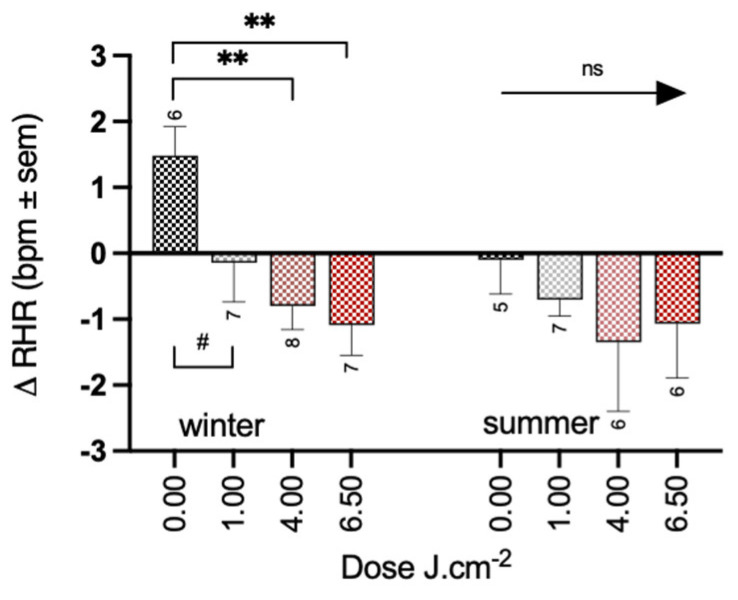
Short-term effects of PBM treatment (average between one stimulation and baseline and two stimulations and baseline) on resting heart rate for summer and winter separately. Significance codes: ** *p* < 0.01, # *p* < 0.1, ns: not significant.

**Table 1 biology-12-00060-t001:** Dosimetry overview. # refers to ‘number of photons’.

	Dose	Duration	Duty Factor		Peak Irradiance	Photonic Dose	Molar Dose
	J·cm^–2^	m	s	%	mW. cm^–2^	# .cm^–2^	µmol. cm^−2^
Low	1.0	42	2520	8%	5	4.3 × 10^18^	7.2
Med	4.0	168	10,080	8%	5	1.7 × 10^19^	28.6
High	6.5	180	10,800	12%	5	2.8 × 10^19^	46.0

**Table 2 biology-12-00060-t002:** Demographics of the individuals in the different groups for summer and winter together. Except for the number of male and female participants, all values are shown as average (standard error of the mean, SEM is noted between brackets). Abbreviations used: PSQI = Pittsburgh Sleep Quality Index, ESS = Epworth Sleepiness Scale, BDI = Beck’s Depression Inventory, BMI = Body mass index, M= Male, F = Female.

	0 J·cm^−2^	1 J·cm^−2^	4 J·cm^−2^	6.5 J·cm^−2^	Significance
Number (M:F)	13 (4:9)	14 (5:9)	15 (7:8)	14 (6:8)	
Age (y)	37.9 (3.4)	38.2 (3.3)	38.7 (5.8)	37.4 (3.5)	ns
Chronotype (h)	4.1 (0.4)	4.2 (0.4)	4.3 (0.4)	4.3 (0.5)	ns
Sleep duration (h)	7.0 (0.4)	7.3 (0.3)	7.4 (0.3)	6.9 (0.3)	ns
Sleep deficit (h)	1.4 (0.4)	1.6 (0.3)	1.0 (0.2)	1.5 (0.5)	ns
PSQI	10.7 (1.0)	10.4 (1.1)	9.8 (0.9)	10.6 (0.7)	ns
ESS	8.6 (1.1)	7.7 (1.5)	8.5 (1.3)	9.6 (1.2)	ns
BDI	10.5 (1.2)	11.1 (1.6)	12.3 (1.4)	11.1 (1.5)	ns
BMI (kg)	25.4 (1.1)	25.3 (1.5)	24.7 (1.1)	24.6 (0.9)	ns

## Data Availability

Not applicable.

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
