# Peer review of "Effects of Near-Infrared Light on Well-Being and Health in Human Subjects with Mild Sleep-Related Complaints: A Double-Blind, Randomized, Placebo-Controlled Study"

_biology, 2022, doi:10.3390/biology12010060_

Round 1

Reviewer 1 Report

In the current study, the authors reported the therapeutic effects of near-infrared light on well-being and health in human subjects with mild sleep-related complaints. This is a double blind, randomized, placebo-controlled clinic study and thus, the results are valuable for the clinical practice. Several issues should be addressed to improve the quality of the report.

1.       1, As a clinical study, the report is suffered from the small sample size, for example, some group only 6 subjects. If the authors run the power study, it would be difficult to obtain the significant difference among the treatments. This important issue should be addressed in the text.

2.      2.  It mentioned that “NIR illumination of the skin of the face, neck, hands was performed on subjects seated behind a desk”. This is not a scientific expression and the real area of illumination should be calculated or estimated (cm2).

    3.  The potential mechanisms of NIR on mitochondrial function should be discussed in details.

     4.    The sample size in each figure and table should be identified.   

Author Response

Dear reviewer.

Thank you for taking the time to have a look at our work. I am happy to read you also see value in these findings.

Please, find below our responses to your comments. I hope you will find them to be satisfactory.

  1. We would like to clarify that sample size as shown in Table 2 are 13, 14, 15 and 14 for the placebo, 1, 4 and 6.5 PBM conditions, respectively. This is indeed reduced to the half when assessing the effects of seasons. The following text has been added to the discussion in which the relatively small sample size is mentioned as a limitation, page 17: 681-685: "

    The ease at which a natural PBM dose can be experienced in summer, might explain the consistency of our stand-out findings occurring only in the winter group, even though the sample size for the season comparison was relatively small and should be considered a limitation on which to improve upon in following studies."

  2. The current study was a field study, with careful instructions. However, we are not able to measure the actual exposure areas of the skin. To support this methodological issue, we have made some calculations in order to specify this better as requested by the reviewer. The following text was added to the device section, page 3: 126-128: “This resulted in an estimated total illuminated area of approximately 1850 cm2 and of 1600 cm2 for male and female respectively, as calculated using https://msis.jsc.nasa.gov/, volume 1, section 3. ”

  3. The potential mechanism of NIR via mitochondrial function is discussed in the discussion section on page 18: 743-749. We have expanded this information as followed: “The selection of this particular group of subjects was premised on the mitochondrial mechanism of PBM. This is the most accepted mechanism of action and it is based on photons being absorbed by cytochromes, which are present inside the mitochondria either as functional proteins or as electron transport shuttles (i.e. cytochrome c oxidase, CCO). This leads to an increased production of ATP, increased oxygen consumption, raised mitochondrial membrane potential, and increased mitochondrial biogenesis have all been shown in vitro after PBM resulting in transient increase in ATP, ROS and NO levels.
  4. Sample size has been added to all figures and tables

Reviewer 2 Report

The scientific paper "Effects of near-infrared light on well-being and health in human subjects with mild sleep-related complaints: a double blind, randomized, placebo-controlled study" aimed to exploits these developments (NIR-LEDs) to explore systemic effects of exposure to 850 nm NIR light in people suffering from mild sleep problems (i.e. sleep deficit and/or reduced sleep quality with clear complaints during daytime) in a double-blind placebo-controlled study.

It can be considered that:

1) The objective of the research must be more explicit in the abstract, as well as its conclusions;

2) On lines 123 and 124, adjust the way references are called (numbers);

3) Insert the manufacturers, city and country of the materials/equipment/products used throughout the text;

4) Does the study have limitations? If so, insert at the end of the discussion;

5) Adjust references according to the standard established by the Biology MDPI journal.

Author Response

Dear reviewer.

Thank you very much for your positive reaction and the helpful comments. Please, find below our responses to your comments. I hope you will find them to be satisfactory.

1. The abstract has been rephrased as followed in order to describe the goal and conclusions of the study clearer:

Modern urban human activities are largely restricted indoors, deprived of direct sunlight containing visible and near-infrared (NIR) wavelengths at high irradiance levels. Therapeutic exposure to doses of red and NIR, known as photobiomodulation (PBM), has been effective for a broad range of conditions. In a double-blind, randomized placebo-controlled study we aimed to assess the effects of a PBM home set-up on various aspects of well-being, health, sleep, and circadian rhythms in healthy human subjects with mild sleep complaints. The effects of three NIR light (850 nm) doses (1, 4, or 6.5 J.cm-2) were examined against placebo. Exposure was presented 5 days per week between 9:30 am and 12:30 pm for 4 consecutive weeks. The study was conducted in summer and winter to include seasonal variation. The results showed PBM treatment only at 6.5 J.cm-2 to have consistent positive benefits on well-being and health, specifically improving mood, reducing drowsiness, reducing IFN- γ, and resting heart rate. This was only observed in winter. No significant effects on sleep or circadian rhythms were noted. This study provides further evidence that adequate exposure to NIR, especially during low sunlight conditions like in the winter, can be beneficial for human health and wellness.

2. The references have been adjusted.

3. Information about the manufacturers, city and country of the materials/equipment/products have been added to the manuscript.

4. The relatively small sample size when assessing the differences between the summer and winter group has been added as a limitation in the discussion section.

5. References have been adjusted.